

# Common eating habit patterns are associated with a high maximum occlusal force and pre-eating cardiac vagal tone

Masahiro Okada[1], Kosuke Okada[2] and Masayuki Kakehashi[3]

[1] Department of Food and Dietetics, Hiroshima Bunka Gakuen Two-Year College, Hiroshima, Japan
[2] Department of Internal Medicine COOP Saeki Hospital, Hiroshima, Japan
[3] Graduate School of Biomedical & Health Sciences, Hiroshima University, Hiroshima, Japan

Corresponding author
Masahiro Okada, okada@hbg.ac.jp

## ABSTRACT

**Background.** Masticatory function is associated with nervous function, including autonomic nervous function, and both functions are influenced by human habits. In a previous preliminary study of 53 young women, we found that eating habit patterns were associated with occlusal force as an indicator of masticatory function. Therefore, we hypothesized that relationships exist between occlusal force, the autonomic nervous system, and eating habit patterns.

**Methods.** To test our hypothesis, we analyzed the relationship between heart rate variability measured before and after lunch in 53 young women, and measured and surveyed maximum occlusal force and eating habit patterns, respectively, in these participants.

**Results.** High occlusal force was associated with an increased high-frequency (HF) component (vagal tone index) of the heart rate variability index immediately before lunch (standardized regression coefficient $(\beta) = 0.417$, $P = 0.002$). Each of the eating habit items surveyed in a questionnaire showed a similar tendency for the HF component immediately before lunch and maximum occlusal force; in particular, "Habit of eating breakfast" and "Number of meals per day" were significantly associated with both variables. Additionally, total eating habit scores related to higher maximum occlusal force were associated with an increase in the HF component immediately before lunch ($\beta = 0.514$, $P < 0.001$). The maximum occlusal force and the pre-eating HF component values were stratified by total eating habit scores (into low, medium, high categories), and the high scores were significantly higher than the medium or low scores.

**Conclusions.** Occlusal force and the pre-eating cardiac vagal response of individuals were characterized by their common eating habit patterns, indicating that eating habits may be simultaneously associated with the development of masticatory function, nervous system development, and cardiovascular rhythm. Although further research is needed to investigate these relationships in detail, our findings provide insights that will inform the study of physical functions, neurodevelopment, habitual behaviors, and health in humans.

## INTRODUCTION

The mastication of food is an essential part of digestion in humans. Occlusal force, *i.e.,* the force exerted on opposing teeth when the jaws are closed or tightened, is a component of masticatory function (*Hatch et al., 2001*) that gradually increases from childhood to adolescence but gradually weakens in older adults (*Bakke et al., 1990*). Therefore, occlusal force is important in the physical development of children and the physical dysfunction and frailty of older adults (*Hirao et al., 2015*; *Iwasaki et al., 2018*). The relationships between occlusal force and the texture and nutrition of food have been studied (*Krall, Hayes & Garcia, 1998*; *Yamanaka et al., 2009*); however, the relationship between occlusal force and masticatory behavior remains unclear (*Sato & Yoshiike, 2011*). The development of occlusal force involves many daily eating habit patterns (*Okada, Okada & Kakehashi, 2022*; *Pedroni-Pereira et al., 2018*), and we found that eating habit patterns may predict maximum occlusal force in young women (*Okada, Okada & Kakehashi, 2022*).

Mastication research has been expanded into the field of neuroscience. Studies have shown a strong relationship between mastication, including occlusal force, and memory and cognitive function (*Weijenberg et al., 2015*; *Takeshita et al., 2016*). Higher brain functions, such as memory and learning, seem to be associated with masticatory function (*Hansson et al., 2013*). In experiments on monkeys and mice, masticatory stimulation has been shown to influence learning, physical development, and mental development (*Rolls, Verhagen & Kadohisa, 2003*; *Nose-Ishibashi et al., 2014*). In addition, masticatory stimuli have been shown to affect neurodevelopment at various ages (*Frota de Almeida et al., 2012*). Moreover, activity in various parts of the brain is involved in regulating occlusal force (*Yoshizawa et al., 2019*).

The relationship between masticatory function and the central nervous system has been studied extensively; however, research on the relationship between masticatory function and autonomic nerve activity according to heart rate variability is lacking. In two separate studies, Ishii et al. revealed the relationship between the masseter muscle and the parasympathetic reflex in rats (*Ishii et al., 2005*) and male–female differences in the cholinergic activity of the parasympathetic vasodilatation (*Ishii, Niioka & Izumi, 2009*). Heart rate variability analysis indicators are used to understand the activity of the human autonomic nervous system (*Malik et al., 1996*), and the sympathetic nerves function predominantly during mastication, *e.g.,* when chewing gum (*Shiba et al., 2002*). In a pilot study, *Takeuchi et al. (2013)* found that impaired masticatory performance may lead to high sympathetic nervous system activity. However, the relationship between masticatory function and the autonomic nervous system requires further clarification; thus, further human-based research on this relationship is required.

The relationship between autonomic nervous balance according to heart rate variability and the eating habits affecting masticatory function has been previously investigated. The eating habits included the eating behaviors, which were focused on, in our preliminary study (*Okada, Okada & Kakehashi, 2022*). For example, *Ozpelit & Ozpelit (2017)* suggested that skipping breakfast may be a cause of cardiac autonomic dysfunction, and *Yoshizaki et al. (2013)* suggested that delaying mealtimes shifts the phase of the circadian rhythm
of the autonomic nervous system. The high-frequency (HF) component of the heart rate variability analysis index reflects vagal tone, and individual differences in the regulation of vagal tone have been studied in relation to not only healthy living but also cognition, behavior, memory, and learning (*Jandackova et al., 2019*; *Thayer et al., 2009*).

Previous studies have linked masticatory function with the autonomic nervous system, both of which may be related to eating habits; however, to the best of our knowledge, the combined relationship among occlusal force, the autonomic nervous system, and human eating habit patterns has not been studied in detail. In the present study, we hypothesized that maximum occlusal force is related to the autonomic nervous system before and after eating and that eating habits related to occlusal force are associated with the autonomic nervous system. To test these hypotheses, we analyzed the relationships among eating habits, occlusal force, and heart rate variability before and after lunch in 53 young and healthy women.

## MATERIALS & METHODS

### Ethics statement

This study is a part of a project on food intake in humans (*Okada & Kakehashi, 2014*). The purpose of this study was to analyze the factors that influence physiological changes before and after food consumption in humans. In this project, physiological factors (including heart rate variability) before and after lunch, physical parameters (including maximum occlusal force), and lifestyle were investigated in 53 healthy young women (project period: March 2010 to April 2012). The study was approved by the Human Studies Ethical Committee of Hiroshima Bunka Gakuen Two-Year College (ethics approval number 22001). The purpose and methods of the study were explained to all participants, after which their oral and written informed consent was obtained.

### Participants and study design

The 53 participants were female Japanese university student volunteers (aged 18–29 years). The present study was conducted after our preliminary study; both involved the same participants and data were collected based on the study design previously described in *Okada, Okada & Kakehashi (2022)*. Each participant completed the study in 1 day on a nonworking day or holiday. All participants were in good health, were sleeping well, had an appetite, were nonsmokers, had not taken any medication, had no chronic migraines, and had no major medical history of cancer, cardiovascular disease, or endocrine disease. The exclusion criteria were as follows: feeling pain or discomfort in the mouth, teeth, face, head, or jaw on the day measurements were taken; prosthodontic therapy, dental braces, excessive weight loss, or hospitalization in the previous 3 months; and/or menstruation during the study period. Participants refrained from physical exercise from breakfast to before lunch. The participants' breakfast content was noted, and we confirmed that they had not drunk alcohol nor consumed caffeine or capsaicin. The participants entered the measurement room 1 h before the measurements were taken to allow adaptation to the room's temperature (mean $\pm$ standard deviation (SD): 20.3 °C $\pm$ 0.9 °C). The participants were weighed and their height was measured while wearing light loungewear with empty

pockets and no shoes, and they answered an eating habits questionnaire before their occlusal force and heart rate variability were measured. These measurements were taken between 11:30 and approximately 13:30, *i.e.,* before and after lunch. Blood samples were also collected from 34 participants before lunch. Twenty minutes after the occlusal force was measured and immediately before eating, heart rate and heart rate variability were measured. The participants confirmed that at least 3 h had passed since breakfast and then ate lunch, which was the same for each participant, *i.e., gyūdon* (beef bowl) consisting of rice and beef (total energy: 3372 kJ; 66.4% carbohydrates, 12.8% protein, 20.8% fat, and 790 mg of sodium). Immediately after lunch and then at 30 min and 1 h after lunch, the heart rate and heart rate variability of the participants was measured again.

Environmental factors that may have affected heart rate variability and masticatory muscle during its measurement were outdoor temperature (mean ± SD: 18.1 °C ± 9.0 °C), atmospheric pressure (1007.1 ± 6.8 hPa), and relative humidity (67.0% ± 10.9%) (*Okada & Kakehashi, 2014*; *Cioffi et al., 2017*). Environmental factor data were obtained from the Hiroshima Local Meteorological Observatory.

## Subjective eating habits questionnaire and maximum occlusal force

The methods used to measure occlusal force, conduct the questionnaire, and analyze the results were partial adopted from our previous study (*Okada, Okada & Kakehashi, 2022*). We used the Okada–Kakehashi questionnaire on eating habits (File S1) consisting of 12 items related to these habits. The total scores for eating habit patterns were calculated using a simple addition method for each item. Occlusal force data were measured using the Dental Prescale system (Fujifilm, Tokyo, Japan) (*Ando et al., 2009*). Each participant sat in a chair without a backrest on a flat floor with a test film set in their mouth; then, they lightly chewed to check the film fit. Each participant then set the measurement film in their mouth and bit it for 3 s at maximum occlusal force. While biting, the head of each participant was held to ensure that the occlusal plane was parallel to the floor. The occlusal force made by each participant was measured twice, and the highest value was considered the maximum occlusal force. Pressure-sensitive film (50H type) was used, and the occlusal force was scanned and quantified using an Occluzer FPD-707 scanner (GC International, Tokyo, Japan). Occlusal force data were expressed in newtons (N), and the force data were recorded as the pressure of the whole dentition for each participant.

## Blood glucose level measurement

Blood samples were taken from 34 participants (19 participants declined to give blood) before lunch to measure plasma glucose levels (mg/dl), which were determined using the glucose oxidase method from finger-prick blood samples (Sanwa Kagaku Kenkyusho Corporation, Nagoya, Japan).

## Heart rate variability measurement

Participants wore light loungewear and sat in a chair when the measurements were taken. A SA-3000P device (Tokyo Iken Corporation, Tokyo, Japan) was used to measure and calculate heart rate variability. Participants were equipped with a hand clip sensor on their index finger and were monitored for 5 min. The heart rate variability power

spectrum was obtained *via* fast Fourier transform analysis of the measured RR interval. Heart rate variability analysis was performed according to the guidelines of the Task Force of the European Society of Cardiology and the North American Society of Pacing and Electrophysiology (*Malik et al., 1996*). The time domain of heart rate variability was indicated by the standard deviation of normal-to-normal RR intervals (SDNN). The frequency domain of heart rate variability analysis was divided into several indicators. Total power was considered the total calculated value of the power spectrum density from frequency 0 to 0.40 Hz in a 5 min measurement. The power spectrum of the frequency band comprises very low frequency (0.0033–0.0400 Hz), low frequency (0.04–0.15 Hz), and HF (0.15–0.40 Hz). The HF component is an indicator of vagal activity and affected by respiratory activity (*Hayano et al., 1994*). The calculated low-to-high-frequency ratio reflects the balance between sympathetic and parasympathetic nervous system activity.

**Data analysis**
All data were analyzed with SPSS for Windows version 24.0 (IBM SPSS, Tokyo, Japan). The sample size was estimated based on previous data (*Okada, Okada & Kakehashi, 2022*) and calculated with a power of 0.80, an effect size equal to eta squared as 0.21 (f-value approximately 0.52), an $\alpha$ level of 0.05, and three groups, considering the ANOVA analysis. In all participants, the characteristics and heart rate variability values were expressed as means $\pm$ standard deviation or means $\pm$ standard error. For checking data normality, the Shapiro–Wilk test was used. Correlations between blood glucose levels before lunch and heart rate variability indices before and after lunch were analyzed using the Pearson's correlation coefficient. Multiple regression analysis was used to analyze the relationships between maximum occlusal force and heart rate variability before and after lunch as well as the relationships among questionnaire score, maximum occlusal force, and heart rate variability before and after lunch. Multivariate analysis was performed by adjusting age, height, weight, room temperature, outdoor temperature, atmospheric pressure, and relative humidity. The total score for eating habits was divided into three levels based on the increase in maximum occlusal force, *i.e.,* low (17–20), medium (21–24), and high (25–28), and multiple comparison analysis of the maximum occlusal force, HF component, and low-to-high-frequency ratio was performed. Multiple comparisons were conducted *via* one-way ANOVA and were Bonferroni adjusted. *P* values of <0.05 (two-tailed) were considered statistically significant. The effect size of regression analysis between maximum occlusal force and HF component before lunch and total eating habit scores were expressed as adjusted R-squared. According to the guide for adjusted R-squared, the small effect size was 0.02, medium effect size was 0.13, and strong effect size was 0.26. The effect size for multiple comparisons by one-way ANOVA was expressed as the observed power. According to the guide for observation power, strong effect size was 0.80 or more.

## RESULTS

Table 1 shows the participants' characteristics ($n = 53$), including their maximum occlusal force, and Table 2 shows the heart rate and heart rate variability index values before and after lunch. No significant correlation was observed between blood glucose levels

**Table 1 Characteristics of participants.**

| Characteristics | n | Mean ± SD | Range |
|---|---|---|---|
| Age (years) | 53 | 20.4 ± 2.6 | 18–29 |
| Height (m) | 53 | 1.6 ± 0.1 | 1.5–1.7 |
| Weight (kg) | 53 | 52.0 ± 7.4 | 41.7–72.6 |
| Blood glucose levels before lunch (mg/dl) | 34 | 89.2 ± 12.9 | 70.0–127.0 |
| Maximum occlusal force (N) | 53 | 686.7 ± 300.8 | 98.0–1578.0 |

**Notes.**
SD, standard deviation; N, newton.

**Table 2 Values of heart rate variability indices before and after lunch.**

| Heart rate variability indices | Immediately before lunch | Immediately after lunch | 30 min after lunch | 1 h after lunch |
|---|---|---|---|---|
| Heart rate (beats/min) | 73.6 ± 10.5 | 78.0 ± 11.5 | 78.8 ± 12.1 | 76.8 ± 11.9 |
| SDNN (ms) | 52.1 ± 17.0 | 52.4 ± 18.5 | 48.5 ± 15.1 | 50.1 ± 16.6 |
| Total power ($ms^2$) | 2355.9 ± 1763.5 | 2362.8 ± 1641.7 | 1953.0 ± 1321.8 | 2075.4 ± 1369.8 |
| Very low frequency ($ms^2$) | 1005.2 ± 878.2 | 780.4 ± 558.9 | 758.4 ± 804.8 | 720.2 ± 654.4 |
| Low frequency ($ms^2$) | 743.6 ± 773.5 | 877.9 ± 815.3 | 594.0 ± 595.0 | 566.0 ± 427.8 |
| High frequency ($ms^2$) | 640.8 ± 538.7 | 704.3 ± 652.7 | 600.6 ± 511.9 | 789.1 ± 793.1 |
| Low-to-high-frequency ratio | 1.7 ± 1.6 | 1.8 ± 2.6 | 1.5 ± 1.5 | 1.2 ± 1.3 |

**Notes.**
Values are given as mean ± standard deviation.
SDNN, standard deviation of normal-to-normal RR intervals.

before lunch in 34 participants and heart rate variability indices Immediately before lunch (SDNN: correlation coefficient (r) = 0.033, $P = 0.852$; Total power: $r = 0.136$, $P = 0.444$; Very low frequency: $r = -0.082$, $P = 0.646$; Low frequency: $r = 0.232$, $P = 0.187$; HF: $r = 0.219$, $P = 0.213$; Low-to-high-frequency ratio: $r = -0.154$, $P = 0.383$). Furthermore, no significant correlations were observed between heart rate variability indices after lunch and blood glucose levels before lunch.

Table 3 shows the relationships between maximum occlusal force and heart rate and heart rate variability before and after lunch. Immediately before lunch, the maximum occlusal force and HF component of heart rate variability were significantly associated (standardized regression coefficient ($\beta$) = 0.417, $P = 0.002$, adjusted R-squared = 0.138), and a higher maximum occlusal force was significantly associated with a decrease in the low-to-high-frequency ratio ($\beta = -0.322$, $P = 0.025$). The maximum occlusal force and HF were significantly associated immediately after ($\beta = 0.308$, $P = 0.035$) and 30 min after ($\beta = 0.343$, $P = 0.013$) lunch.

Table 4 shows the associations between the 12 eating habit–related questionnaire items, maximum occlusal force, and HF components before and after lunch. "Habit of eating breakfast" was significantly associated with an increase in maximum occlusal force ($\beta = 0.384$, $P = 0.008$) and the HF component immediately before lunch ($\beta = 0.363$, $P = 0.009$). "Number of meals per day" was significantly associated with maximum occlusal force ($\beta = 0.356$, $P = 0.014$) and the HF component immediately before lunch ($\beta = 0.307$,
**Table 3** Relationship between maximum occlusal force and heart rate variability before and after lunch.

| Heart rate variability indices | Immediately before lunch $\beta$ (*P* value) | Immediately after lunch $\beta$ (*P* value) | 30 min after lunch $\beta$ (*P* value) | 1 h after lunch $\beta$ (*P* value) |
|---|---|---|---|---|
| Heart rate (beats/min) | −0.186 (0.209) | −0.080 (0.599) | −0.109 (0.471) | −0.025 (0.871) |
| SDNN (ms) | 0.176 (0.235) | 0.120 (0.431) | 0.105 (0.493) | −0.066 (0.660) |
| Total power (ms$^2$) | 0.127 (0.398) | 0.175 (0.236) | −0.016 (0.917) | −0.087 (0.568) |
| Very low frequency (ms$^2$) | −0.053 (0.728) | 0.089 (0.539) | −0.258 (0.082) | −0.184 (0.217) |
| Low frequency (ms$^2$) | 0.044 (0.764) | 0.044 (0.772) | 0.018 (0.908) | −0.112 (0.451) |
| High frequency (ms$^2$) | 0.417 (0.002) | 0.308 (0.035) | 0.343 (0.013) | 0.062 (0.645) |
| Low-to-high-frequency ratio | −0.322 (0.025) | −0.196 (0.192) | −0.125 (0.343) | −0.119 (0.351) |

**Notes.**
Multiple regression analysis was conducted after adjusting for age, height, weight, room temperature, outdoor temperature, atmospheric pressure, and relative humidity.
$\beta$, standardized regression coefficient.

$P = 0.027$). For the item "Eating speed," significant relationships existed between fast eaters and the HF component after lunch (immediately after lunch: $\beta = 0.346$, $P = 0.022$; 30 min after lunch: $\beta = 0.293$, $P = 0.043$; 1 h after lunch: $\beta = 0.301$, $P = 0.028$). "Eating until full" and "Eat for stress relief" were significantly associated with maximum occlusal force (Eating until full: $\beta = -0.417$, $P = 0.006$; Eat for stress relief: $\beta = -0.359$, $P = 0.026$), and the HF component immediately before lunch exhibited the same tendency, although not at a significant level. For the item "Eat with others or alone," eating with others was significantly associated with an increase in the HF component immediately before lunch ($\beta = 0.324$, $P = 0.030$).

As shown in Fig. S1, the total eating habit scores based on increasing maximum occlusal force exhibited an almost normal distribution. Table 4 shows the relationships among the total eating habit scores, maximum occlusal force, and HF component before and after lunch. Higher total eating habit scores were significantly associated with an increase in the HF component immediately before lunch ($\beta = 0.514$, $P < 0.001$, adjusted R-squared = 0.261) and an increase in maximum occlusal force ($\beta = 0.612$, $P < 0.001$, adjusted R-squared = 0.266). Figure S2 shows a scatterplot of maximum occlusal force against the total eating habit scores with a simple regression trend line. Figure 1 shows the average HF component immediately before lunch and the frequency of each total eating habit scores. The simple regression trend line was used to represent the increase in HF component with increasing total eating habit scores (High-frequency component = 70.686*X–960.971, $P < 0.001$).

The maximum occlusal force, HF component, and low-to-high-frequency ratio were stratified by the total eating habit scores into low (17–20), medium (21–24), and high (25–28) categories. The high scores were significantly higher than the medium or low scores for both the maximum occlusal force (high *vs.* medium, $P = 0.028$; high *vs.* low, $P < 0.001$; observed power = 0.940; Fig. 2) and pre-eating HF component (high *vs.* medium, $P < 0.001$; high *vs.* low, $P < 0.001$; observed power = 0.970; Fig. 3). For the

Okada et al. (2023), *PeerJ*, DOI 10.7717/peerj.15091

**Table 4** Relationships among eating habits, maximum occlusal force, and the high-frequency component of heart rate variability before and after lunch.

| Eating habit items | Scores | Maximum occlusal force β (P value) | HF immediately before lunch β (P value) | HF immediately after lunch β (P value) | HF 30 min after lunch β (P value) | HF 1 h after lunch β (P value) |
|---|---|---|---|---|---|---|
| 1. Habit of eating breakfast | 1 or 2 | 0.384 (0.008) | 0.363 (0.009) | 0.263 (0.073) | 0.165 (0.244) | 0.018 (0.892) |
| 2. Always eat at a fixed time | 1 or 2 | −0.173 (0.291) | 0.077 (0.622) | 0.160 (0.323) | 0.111 (0.472) | 0.256 (0.075) |
| 3. Number of meals per day (including snacks) | 2–5 | 0.356 (0.014) | 0.307 (0.027) | 0.231 (0.112) | 0.144 (0.303) | 0.008 (0.952) |
| 4. Amount eaten | 1–3 | 0.138 (0.358) | 0.178 (0.214) | 0.120 (0.419) | 0.023 (0.869) | −0.053 (0.696) |
| 5. Eating speed | 1 or 2 | 0.113 (0.472) | 0.245 (0.099) | 0.346 (0.022) | 0.293 (0.043) | 0.301 (0.028) |
| 6. Chew food well | 1 or 2 | 0.017 (0.910) | 0.001 (0.993) | 0.036 (0.810) | 0.020 (0.891) | 0.054 (0.694) |
| 7. Eat until full | 1 or 2 | −0.417 (0.006) | −0.290 (0.051) | −0.046 (0.771) | −0.132 (0.373) | 0.012 (0.933) |
| 8. Think about balance of meal (nutritional balance) | 1 or 2 | −0.239 (0.116) | −0.142 (0.333) | −0.056 (0.713) | 0.025 (0.863) | 0.144 (0.294) |
| 9. Many likes and dislikes | 1 or 2 | −0.003 (0.986) | 0.087 (0.547) | 0.063 (0.673) | 0.051 (0.719) | 0.000 (0.999) |
| 10. Eat for stress relief | 1 or 2 | −0.359 (0.026) | −0.241 (0.123) | −0.132 (0.419) | 0.020 (0.898) | 0.175 (0.233) |
| 11. Eat with others or alone (including family) | 1–3 | 0.186 (0.242) | 0.324 (0.030) | 0.175 (0.265) | 0.277 (0.061) | 0.080 (0.576) |
| 12. Conversation when eating | 1–3 | 0.259 (0.118) | 0.234 (0.139) | −0.034 (0.838) | 0.133 (0.398) | 0.100 (0.505) |
| Total eating habit scores | 17–28 | 0.612 (<0.001) | 0.514 (<0.001) | 0.286 (0.076) | 0.240 (0.120) | −0.004 (0.977) |

**Notes.**

Multiple regression analysis was performed after adjusting for age, height, weight, room temperature, outdoor temperature, atmospheric pressure, and relative humidity.

Total eating habit scores were calculated using simple addition for each item point based on the tendency for increasing maximum occlusal force.

β, standardized regression coefficient; HF, high-frequency.

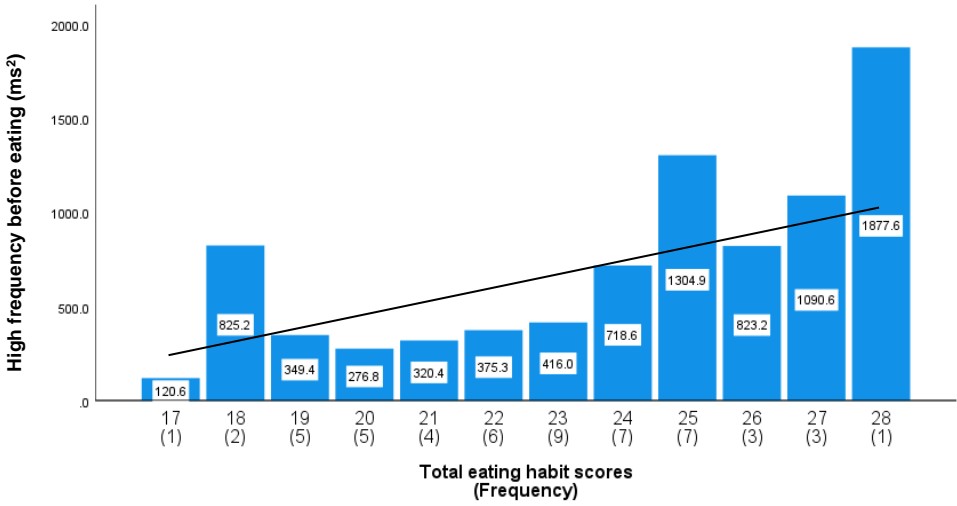

**Figure 1** **The average high-frequency components before lunch by total eating habit scores.** Total eating habit scores from 17 to 28 used scores related to increase maximum occlusal force. The number in parentheses indicates the frequency. The values in the bar graph are mean values of the high-frequency components immediately before lunch. Mean values were adjusted for age, height, weight, room temperature, outdoor temperature, atmospheric pressure, and relative humidity. The regression trend line is shown as a solid line (High-frequency component = 70.686*X–960.971, $P < 0.001$).

pre-eating low-to-high-frequency ratio, the low scores were significantly higher than the medium or high scores (low *vs.* medium, $P < 0.01$; low *vs.* high, $P < 0.001$; observed power = 0.948; Fig. S3).

## DISCUSSION

Occlusal force, which is a major component of masticatory function (*Hatch et al., 2001*), is influenced by age and sex; thus, it is important to consider the age and sex of the participants when designing a study on occlusal force (*Bakke et al., 1990*). We selected women aged 18–29 years who had a stable maximum occlusal force as participants. Masticatory function is thought to be closely related to the central nervous system (*Hansson et al., 2013*; *Rolls, Verhagen & Kadohisa, 2003*; *Nose-Ishibashi et al., 2014*; *Frota de Almeida et al., 2012*; *Yoshizawa et al., 2019*); however, the relationship between occlusal force and autonomic nervous activity before and after food intake has rarely been studied. Masticatory function has been associated with autonomic nervous balance in subjects with a wide age range (*Takeuchi et al., 2013*). The autonomic nervous balance measured before and after lunch in the present study is considered to represent a normal change, even when considering the low-to-high-frequency ratio values (*Okada & Kakehashi, 2014*; *Harthoorn & Dransfield, 2008*). The potential for maintaining sympathetic dominance during food mastication also exists, although the response is lower than that measured during the chewing of gum (*Shiba et al., 2002*). In the present study, we showed that an increase in maximum occlusal force was associated with an increase in the HF component and parasympathetic dominance, particularly immediately before eating lunch. In patients

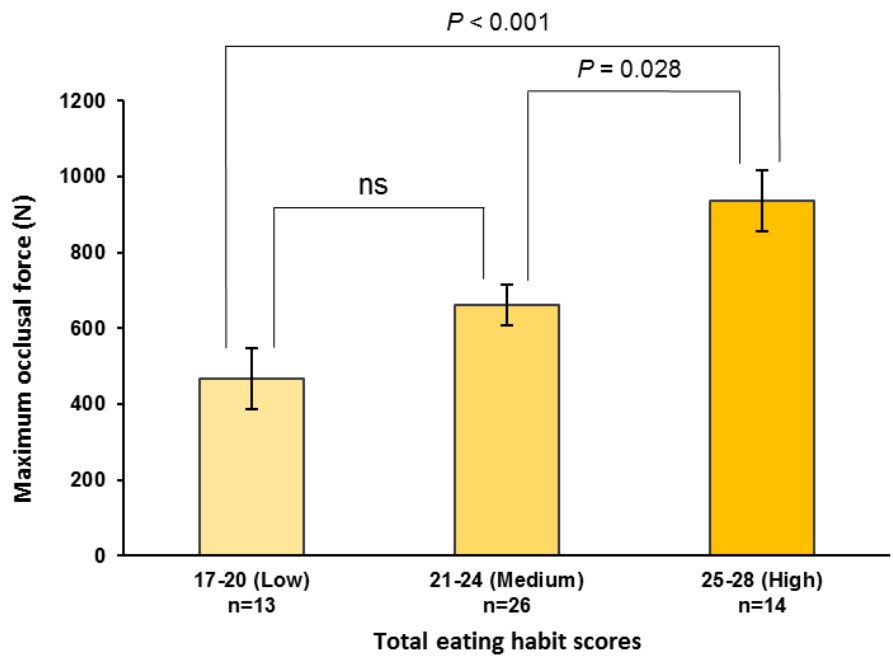

**Figure 2 Maximum occlusal force stratified by total eating habit scores.** Maximum occlusal force is stratified by total eating habit scores, *i.e.*, low (17–20), medium (21–24), and high (25–28) scores. One-way ANOVA with Bonferroni adjustment was used for multiple comparisons after adjusting for age, height, weight, room temperature, outdoor temperature, atmospheric pressure, and relative humidity. Values are means ± standard error. ns, not significant.

with migraine, clenching has also been shown to enhance vagal tone (*Zaproudina et al., 2018*). The present results are considered to represent a normal pre-eating vagal response given that the participants did not experience migraine and a sufficient interval was used between occlusal force measurements. Thus, we suggest that a relationship exists between the development of occlusal force and the vagal response before food intake.

Importantly, the eating habit patterns associated with high occlusal force were also associated with an increase in the pre-eating HF component of heart rate variability. Eating behavior (including breakfast skipping), appetite, and social factors (loneliness) affect the parasympathetic nervous system and vagal tone (*Ozpelit & Ozpelit, 2017*; *Yoshizaki et al., 2013*; *Wilson et al., 2019*; *González-Velázquez et al., 2020*). Thus, these may be common factors in mastication, occlusal force, and pre-eating vagal tone. A relationship between respiration and the HF component of heart rate variability may also exist (*Hayano et al., 1994*). In our study, "Eating speed" was significantly related to the HF component only after eating. The speed of masticatory movement may affect respiratory activity during daily eating. In a task involving the imagination of dynamic exercise, respiratory responses were shown to increase depending on the image presented (*Wuyam et al., 1995*), and respiratory responses may be affected by learning effects, including exercise habits (*Thornton et al., 2001*). Thus, the effect of respiration on the increase in the pre-eating HF component must

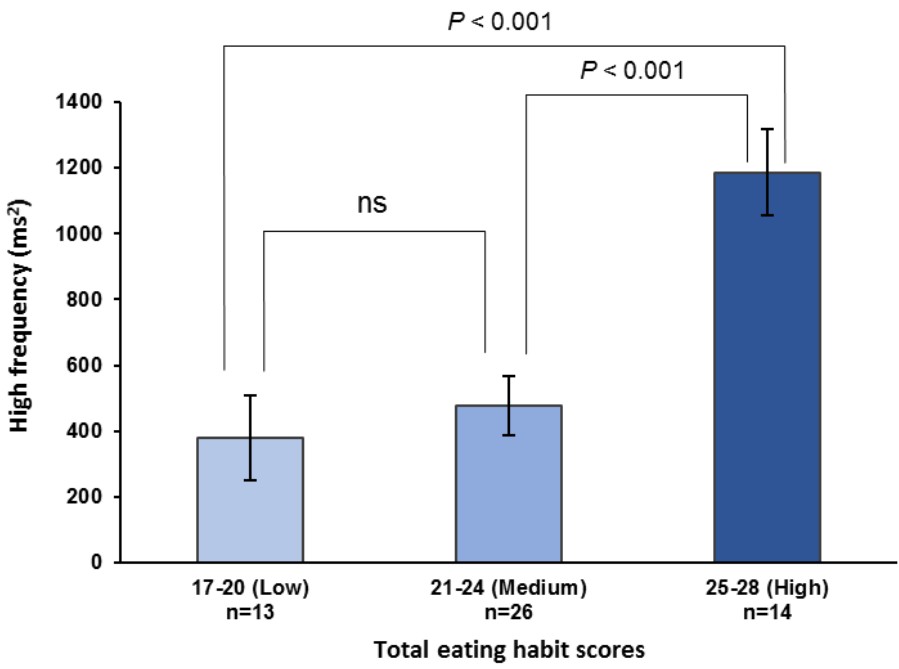

**Figure 3  High-frequency components before lunch stratified by total eating habit scores.** High-frequency components immediately before lunch are stratified by total eating habit scores based on an increase in maximum occlusal force, *i.e.,* low (17–20), medium (21–24), and high (25–28) scores. One-way ANOVA with Bonferroni adjustment was used for multiple comparisons after adjusting for age, height, weight, room temperature, outdoor temperature, atmospheric pressure, and relative humidity. Values are means ± standard error. ns, not significant.

be considered, and further assessment of respiration and the vagal response in relation to food intake is required.

Pre-eating vagal responses may be affected by habit patterns. During the performance of sport, the self-regulation of vagal activity is important to psychophysiology and has been discussed in relation to a variety of behavioral and environmental factors (*Laborde, Mosley & Ueberholz, 2018*). Higher vagal tone may also favor higher brain functions in the context of behavior (*e.g.,* better executive cognitive performance, emotional and health regulation, and social functioning) (*Laborde, Mosley & Thayer, 2017*). The aforementioned factors include both short-term and long-term habits, but a long-term perspective may be required in relation to physical development and vagal activity. Occlusal force is important in sports performance (*Sannajust et al., 2002*), and increases in occlusal force may not exhibit long-term stability with only short-term training (*Ohira et al., 2012*). Occlusal force has been found to gradually and steadily increase and stabilize over approximately 20 years (*Bakke et al., 1990*). In mice, early growth is an important factor in mastication and nerve development (*Nose-Ishibashi et al., 2014*; *Frota de Almeida et al., 2012*). In humans, the development of vagal activity begins early in childhood (*Pivik et al., 2015*). Therefore, the pre-eating vagal response of an individual might be acquired in association with long-term habits in a similar manner to occlusal force.
It is important to consider how eating habits affect occlusal force and the nervous system. Mastication ability, including occlusal force, is closely related to the nervous system and its functions, including cognition and memory (*Weijenberg et al., 2015*; *Takeshita et al., 2016*; *Hansson et al., 2013*). The habits involved in developing occlusal force may directly contribute to neurodevelopment related to cognitive behavior. During food intake, the nervous system responds to images and environmental factors (*e.g.*, social contact and loneliness) (*Wilson et al., 2019*; *González-Velázquez et al., 2020*; *Takada et al., 2018*). Therefore, a network through the midbrain may control human cognitive behavior (*Wei, Chen & Wu, 2018*). If the pathway from pre-eating to mastication is considered a cognitive performance, the enhanced vagal tone before eating might be important (*Thayer et al., 2009*; *Laborde, Mosley & Thayer, 2017*). Therefore, some eating habits may be directly related to the vagal response before eating. Our questionnaire, which included habitual items that covered both occlusal force and vagal activity, may have helped clarify the relationship. Before eating, many stimuli, cognitive tasks, reactions, and behaviors are concentrated; hence, this period may be an important hotspot for understanding the nervous system and physical reactions. Individual eating habits include many stimuli that directly or indirectly (through mastication) affect the development of autonomic nervous responses, and an individual's underlying autonomic nervous response may be characterized by habit-induced nervous system development.

During human physical and nervous system development, *e.g.*, the development of occlusal force and vagal nerve activity, respectively, many eating habits are combined. In middle-aged and older people, decreases in physical activity (*e.g.*, occlusal force) and vagal nerve activity may occur simultaneously, which may be an important issue for healthy living and mortality (*Malik et al., 1996*; *Jandackova et al., 2019*). Indeed, eating habit patterns may be an important factor in maintaining physical and neural activity in all phases of life.

This study had some limitations. First, the sample size was small (only 53 young women). Both occlusal force and heart rate variability indices are affected by sex and age (*Bakke et al., 1990*); therefore, a larger sample consisting of male and female participants with a wider age range is required to further investigate eating habits and autonomic nervous responses. Second, strong restrictions were not placed on the breakfast received by the participants. Given that fasting blood glucose levels affect heart rate variability (*Lutfi & Elhakeem, 2016*), we collected optional blood glucose level data before taking lunch-related measurements. We found no correlation between fasting blood glucose levels and heart rate variability indices. However, other plasma nutrients and hormone secretions, such as insulin, could also be considered as confounding factors. Third, eating habits were assessed before and after lunch. It would be desirable to extend the research to include other meals throughout the day. Fourth, the questionnaire used had a simple design. An improved questionnaire that categorizes eating habits, considering exercise and sleep habits, and evaluates both mastication and neurodevelopment might provide more useful data. Finally, this was an observational study, and further investigation is required to determine the mechanism underlying the observed phenomena. Despite the limitations of the present study, the results provide important insights into the links between eating habits, masticatory ability, and autonomic nervous system responses in humans.

## CONCLUSIONS

We hypothesized that maximum occlusal force is related to the autonomic nervous system before and after eating and that eating habits related to occlusal force are associated with the autonomic nervous system. We found that the occlusal force and pre-eating vagal response of individuals can be characterized by their common eating habit patterns, suggesting that eating habits may be simultaneously associated with the development of human masticatory function, nervous system development, and cardiovascular rhythm. Overall, these findings will inform further research into the physical functions, neurodevelopment, habitual behaviors, and health of humans.

## ACKNOWLEDGEMENTS

We thank Hiroshima Bunka Gakuen University for lending us the analytical equipment and the staff of Hiroshima University School of Dentistry for providing appropriate advice related to our research.

### Funding

The authors received no specific funding for this work. The funders had no role in study design, data collection and analysis, decision to publish, or preparation of the manuscript.

### Competing Interests

The authors declare there are no competing interests.

### Author Contributions

- Masahiro Okada conceived and designed the experiments, performed the experiments, analyzed the data, prepared figures and/or tables, authored or reviewed drafts of the article, and approved the final draft.
- Kosuke Okada performed the experiments, authored or reviewed drafts of the article, and approved the final draft.
- Masayuki Kakehashi conceived and designed the experiments, authored or reviewed drafts of the article, and approved the final draft.

### Human Ethics

The following information was supplied relating to ethical approvals (i.e., approving body and any reference numbers):

The study was approved by the Human Studies Ethical Committee of Hiroshima Bunka Gakuen Two-Year College.

### Data Availability

The raw dataset is available in the Supplemental File.

## Supplemental Information

Supplemental information for this article can be found online at http://dx.doi.org/10.7717/peerj.15091#supplemental-information.

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
