# Peer review of "Common eating habit patterns are associated with a high maximum occlusal force and pre-eating cardiac vagal tone"

_PeerJ, doi:10.7717/peerj.15091_

## Round 0.1 · original submission · Minor Revisions

Congratulations on a good paper; please resolve the second reviewer's remarks

·

Basic reporting

NO COMMENT

Experimental design

NO COMMENT

Validity of the findings

NO COMMENT

Additional comments

The topic is very interesting and innovative. It was valuable finding that the occlusal force corresponds to the cardiac vagal tone. Congratulations for the topic and the excluding creteria.

·

Basic reporting

Dear authors, the English used in the description of the study is clear and professional.
yours is an interesting topic and the recent literature is getting richer with studies analyzing the physiology of the TMJ system in relation to other functions of the central nervous system such as proprioception and autonomic nervous system activities. reading the literature reference list of your article about 28 % of the bibliographic references are older than 15 years and about 23 % are 10 years old. would it be possible to replace these old references with more modern ones given the current large scientific production?
The article structure is conform to the standard sections

Experimental design

The wok is conform to the aims and scope of the journal, with a clear research question.
The introduction is well written. It explains the topic thoroughly. However, when you first introduce the concept of eating habits (line 79) specify what you mean because physiologically from a scientific point of view (inserting references if necessary) they can affect the masticatory function.
Methods are described with sufficient detail however more information are required to be replicable:
- line 114: were there any subjects who had dental braces? please specify this subjects characteristic
- line 139: what kinds of habits ? meal time, quantity of food, quality, etc.? can you put categories to explain better?
- line 145: can you describe the directions you gave the subject to assess occlusal force (was the subject sitting or standing, whether the time of reaching maximum contraction was fixed, etc.? what was the protocol?
- line 175: have you analyzed the normality of the data? with which test?
- line 190: How did you calculate the sample size needed for this study?

Validity of the findings

The work has good innovativeness and can contribute to enrich the literature. However, the practical implications should be further explored.

·

Basic reporting

The article is clear, unambiguous and written in professional English throughout. The introduction and bibliographical references allow to put the article in the broader scientific context. The specifications of PeerJ regarding article structure were taken into account. The figures clarify the main study results and the raw data is available.

Experimental design

The article fits in the subject area of PeerJ. The authors have disambiguated the article from the previously published article. A clearly definable and new hypothesis was formulated. The experiment is described in a comprehensible way.

Validity of the findings

The presented experiment combines eating habits, oral occlusal force and heart rate variability. The questionnaire, measuring devices and statistic methods are documented. The conclusions drawn support the proposed hypothesis.

Additional comments

Thank you for providing this article and presenting your study findings, which offer to connect masticatory function, eating habits and autonomic nervous system. I propose to publish the article as is.

---

## Round 0.2 · accepted · Accept

Congratulations on the excellent paper!